# How Well Are Hand Hygiene Practices and Promotion Implemented in Sierra Leone? A Cross-Sectional Study in 13 Public Hospitals

**DOI:** 10.3390/ijerph19073787

**Published:** 2022-03-23

**Authors:** Sulaiman Lakoh, Anna Maruta, Christiana Kallon, Gibrilla F. Deen, James B. W. Russell, Bobson Derrick Fofanah, Ibrahim Franklyn Kamara, Joseph Sam Kanu, Dauda Kamara, Bailah Molleh, Olukemi Adekanmbi, Simon Tavernor, Jamie Guth, Karuna D. Sagili, Ewan Wilkinson

**Affiliations:** 1Department of Medicine, College of Medicine and Allied Health Sciences, University of Sierra Leone, Freetown, Sierra Leone; gibrilladeen1960@yahoo.com (G.F.D.); samjokanu@yahoo.com (J.S.K.); james.russell@usl.edu.sl (J.B.W.R.); 2Department of Medicine, University of Sierra Leone Teaching Hospitals Complex, Freetown, Sierra Leone; 3Sustainable Health Systems Sierra Leone, Freetown, Sierra Leone; bmollehshs@gmail.com; 4World Health Organization Country Office, Freetown, Sierra Leone; marutaa@who.int (A.M.); derrickfbob@gmail.com (B.D.F.); ikamara@who.int (I.F.K.); 5Ministry of Health and Sanitation, Government of Sierra Leone, Freetown, Sierra Leone; christy.conteh@yahoo.com (C.K.); daudakamara50@gmail.com (D.K.); 6Department of Medicine, University of Ibadan, Ibadan 200005, Nigeria; kemiosinusi@gmail.com; 7Department of Medicine, University College Hospital, Ibadan 200005, Nigeria; 8School of Medicine, University of Liverpool, Cedar House, Ashton Street, Liverpool L69 3GE, UK; s.j.tavernor@doctors.org.uk; 9Global Health Connections, Center Barnstead, Barnstead, NH 03225, USA; guth.jamie@gmail.com; 10International Union Against Tuberculosis and Lung Disease (The Union), Southeast Asia Office, New Delhi 110016, India; ksagili@theunion.org; 11Institute of Medicine, University of Chester, Countess Way, Chester CH2 1BR, UK; e.wilkinson@chester.ac.uk

**Keywords:** healthcare-associated infections (HAIs), hand hygiene self-assessment framework (HHSAF), infection prevention and control, structured operational research initiative training (SORT IT), hand hygiene training, IPC in hospital

## Abstract

Healthcare-associated infections (HAIs) result in millions of avoidable deaths or prolonged lengths of stay in hospitals and cause huge economic loss to health systems and communities. Primarily, HAIs spread through the hands of healthcare workers, so improving hand hygiene can reduce their spread. We evaluated hand hygiene practices and promotion across 13 public health hospitals (six secondary and seven tertiary hospitals) in the Western Area of Sierra Leone in a cross-sectional study using the WHO hand hygiene self-Assessment framework in May 2021. The mean score for all hospitals was 273 ± 46, indicating an intermediate level of hand hygiene. Nine hospitals achieved an intermediate level and four a basic level. More secondary hospitals 5 (83%) were at the intermediate level, compared to tertiary hospitals 4 (57%). Tertiary hospitals were poorly rated in the reminders in workplace and institutional safety climate domains but excelled in training and education. Lack of budgets to support hand hygiene implementation is a priority gap underlying this poor performance. These gaps hinder hand hygiene practice and promotion, contributing to the continued spread of HAIs. Enhancing the distribution of hand hygiene resources and encouraging an embedded culture of hand hygiene practice in hospitals will reduce HAIs.

## 1. Introduction

Healthcare-associated infections (HAIs) result in millions of avoidable deaths or prolonged lengths of stay in hospitals across the world [1,2]. They increase healthcare costs for health systems as well as for the patient’s families and facilitate the spread of multidrug resistant pathogens. The estimated economic loss due to HAIs in high-income countries was about $7 billion; however, such data is not available for low- and middle-income countries (LMICs) [1,3]. Therefore, preventing HAIs is an important strategic intervention for patients’ safety, improving the quality of healthcare, and reducing costs [4].

The HAIs rate in LMICs is at least three times that of high-income countries, especially in Africa, although the incidence of HAIs and the associated costs are still unknown or underestimated [4]. Studies from countries in sub-Saharan Africa have reported high prevalence and incidence rates of HAIs [4,5]. A longitudinal study in an Ethiopian hospital reported an incidence rate of HAIs of 28 per 1000 patient days and a prevalence rate of 19% among inpatients [6]. Complicating the issue is the high incidence of multidrug resistant pathogens in patients with HAIs. Studies conducted in Sierra Leone reported high prevalence rates of extended-spectrum beta-lactamase-producing Gram negative bacteria in patients with healthcare-associated infections [7,8].

In healthcare settings, the hands of healthcare workers play a critical role in the spread of HAI, including multidrug resistant infections. Numerous studies have shown that improved hand hygiene among healthcare workers can reduce the spread of these infections [4,9]. Hence, many hospitals in LMICs are implementing hand hygiene programs [4]. The World Health Organization (WHO) has recognised hand hygiene as one of the core indicators of hospital safety and quality of care. It has developed a simple standardized hand hygiene self-assessment framework (HHSAF) tool, which has been validated in 19 countries, to assess the level of a hospital’s implementation of hand hygiene practice and promotion. This tool identifies gaps and facilitates the development of action plans to improve hand hygiene practices and promotion activities [10,11]. The tool assesses a facility’s practice and promotion of hand hygiene in five domains: system change, training and education, evaluation and feedback, reminders in the workplace, and the institutional safety climate for hand hygiene. It provides a scoring system to determine the level of implementation of hand hygiene [10].

In 2014, Sierra Leone faced an Ebola outbreak that infected over 14,000 people, killed 3589 people, and adversely affected the operations of the health system [12]. As a result, the Ministry of Health established the National Infection Prevention and Control (IPC) Program in 2015 for the first time and developed an IPC policy to guide the implementation of hand hygiene [13]. Since then, the National IPC Program has used the WHO HHSAF tool annually to collect data on the status of hand hygiene promotion activities within healthcare facilities. However, this data has not been routinely analysed.

Analysing the HHSAF data is an important exercise to improve hand hygiene practices and promotion activities in Sierra Leone. Currently, there is a huge gap in knowledge regarding hand hygiene practices and promotion activities owing to the limited utilization of data collected using the HHSAF tool. This reduces the opportunities to improve hand hygiene practices and promotion in the country. Therefore, we carried out this study to evaluate the implementation of the hand hygiene practices and promotion activities in 13 public health hospitals in the Western Area of Sierra Leone using the HHSAF tool to identify specific areas that need improvement in health facilities.

## 2. Materials and Methods

### 2.1. Study Design

This is a cross-sectional descriptive study involving primary data collection.

### 2.2. Study Setting

General setting: Sierra Leone has a population of 7.6 million, of which 42% are under 15 years of age [14]. It is a relatively poor country. In 2018, Sierra Leone’s income per capita was estimated at $470, and agriculture contributed 60% of its gross domestic product [15,16]. The country is divided into five geopolitical regions, one of which is the Western Area (urban and rural). This is the most densely populated region of Sierra Leone with a population of about 1.5 million and includes the capital, Freetown [17].

Specific Setting: Sierra Leone has a three-tiered public health system with primary healthcare as the first level of care, which includes peripheral health units. District hospitals provide secondary care, and regional/national hospitals provide tertiary care. There are 25 public hospitals and 1160 peripheral health units [16].

The Western Area has 13 of the 25 public hospitals as shown in Figure 1. Of these 13 facilities, six are secondary hospitals, and the remaining seven are tertiary hospitals. Each secondary hospital has between 30 and 57 beds, while the tertiary hospitals have between 100 and 300 beds.

Each hospital has an institutional IPC focal person, who is normally a nurse, providing technical leadership for IPC practices. Every ward or unit in the hospital has an IPC link nurse who is responsible for overseeing the IPC practice in their area. The data for the HHSAF is collected annually from the IPC focal person.

### 2.3. Study Population

The study population included the IPC focal persons of the 13 public hospitals within the Western Area urban and rural districts of Sierra Leone.

### 2.4. Data Variables

The general variables collected were the type of hospital, bed capacity, districts where the hospitals are located (urban vs. rural), and the staff capacity (doctors, nurses, community health officers, pharmacy personnel, laboratory personnel, administrators, radiographers, and auxiliary staff).

The specific variables were collected using the WHO HHSAF tool. The hospitals were scored based on the HHSAF scoring criteria depending on the cumulative scores as shown in Table 1 [10,18]. A facility that reached the advanced level rating was asked 20 additional questions under the leadership domain [10].

### 2.5. Data Collection, Analysis, and Statistics

The data was collected by interviewing the IPC focal persons of the 13 public hospitals in May 2021 by the principal investigator and three research assistants using a paper format of the HHSAF tool and transcribing the results onto it.

The collected data was then double entered into EpiData Entry Software (version 3.1 for data collection, EpiData Association, Odense, Denmark), validated, and analysed using the EpiData Analysis software (EpiData Association, Odense, Denmark).

The characteristics of the 13 hospitals were described using the general variables collected. The aggregate HHSAF score for each domain of the tool was calculated, and the total score for each facility was generated. The mean and standard deviation of the HHSAF scores of the health facilities were used to summarize the data. Key indicators under each domain were analysed using simple proportions. The overall data were analysed separately for secondary and tertiary hospitals.

## 3. Results

### 3.1. Characteristics of the Hospitals

As shown in Table 2, seven of the 13 public hospitals selected for this study were tertiary hospitals, located in the Western Area Urban. The mean bed capacity in tertiary hospitals was 168 and that of secondary hospitals was 53. The mean staff capacity in tertiary hospitals was 353, and in secondary hospitals it was 200. In all the hospitals, there were more nurses than other cadres of health service providers as shown in Table 3.

### 3.2. HHSAF Assessment Scores of the Hospitals

#### 3.2.1. The Total Facility Score Generated from the HHSAF Tool

Table 4 shows the scores for the five domains in the HHSAF tool. The mean score (±standard deviation) for the 13 public hospitals in the Western Area was 273 ± 46, indicating an overall intermediate hand hygiene level. The minimum and maximum scores for the 13 hospitals were 210 (basic) and 375 (intermediate), respectively. The overall hospital hand hygiene levels achieved were either basic 4 (31%) or intermediate 9 (69%). More secondary hospitals achieved an intermediate level of hand hygiene than tertiary hospitals 5 (83%) vs. 4 (57%). The mean hand hygiene score in tertiary hospitals (284 ± 58) was higher than those in secondary hospitals (260 ± 67), though both were at the intermediate level.

#### 3.2.2. The Domains Scores of the 2021 HHSAF

The scores for each domain are highlighted in Table 4, and the details of the HHSAF some hand hygiene indicators in each domain are provided in Table A1.

For ease of understanding, visualisation, and action, we have colour-coded the scores of each domain into four categories as shown in Table 4.

#### 3.2.3. The Performance of Hospitals across the Domains

System Change: This domain assessed the availability of hand hygiene resources (alcohol-based hand rub, soap, water and single-use towels, and sink: bed ratio) and systems (dedicated budget and a realistic plan to improve on hand hygiene infrastructure). All 13 hospitals offered alcohol-based hand rub, but only ten and eight hospitals, respectively, reported a continuous supply of soap and running water. None of the hospitals had single-use towels for hand drying. None of the hospitals had a dedicated hand hygiene budget, nor did they have a realistic plan to improve hand hygiene infrastructure.

The range of scores for the system change domain was 35 to 80. Secondary hospitals (48 ± 11) had lower mean scores in the system change domain than tertiary hospitals (56 ± 21). While the performance for five hospitals (three secondary, two tertiary) was rated good, three hospitals (two secondary, one tertiary) were rated excellent in this domain.

Training and Education: This domain assessed the skills of health professionals to conduct hand hygiene training and the systems and processes that supported this training. All 13 hospitals provided some form of training for healthcare workers, and 10 of them had systems that confirm that training has been conducted in the facilities. However, 12 hospitals had no dedicated budget for hand hygiene training. The mean training and evaluation score was slightly higher in secondary hospitals (72 ± 21) than tertiary hospitals (66 ± 13). This was the only domain where most hospitals (92%) had good (three secondary, three tertiary) or excellent (two secondary, four tertiary) scores.

Evaluation and Feedback: Among the indicators assessed on the evaluation and feedback domain were audit, direct and indirect monitoring of hand hygiene compliance, and immediate systematic feedback on hand hygiene.

Regular ward audits of hand hygiene resources were conducted at only two hospitals. Knowledge of hand hygiene indications and correct hand hygiene techniques was assessed among healthcare workers at 11 and 10 hospitals, respectively. In nine hospitals, hand hygiene compliance was indirectly monitored by regular assessment of alcohol-based hand rub and soap consumption. Eleven hospitals conducted direct monitoring of hand hygiene compliance at least every three months, using the WHO hand hygiene observation tool, and three hospitals had compliance rates above 50%. The performance of five of the hospitals (two secondary, three tertiary) in this domain was rated poor.

Reminders in the Workplace: This domain assessed the use of hand hygiene promotional materials, including posters, campaign screensavers, badges, and leaflets. Posters promoting hand hygiene indications, correct hand washing techniques, and the use of alcohol-based hand rub were posted in all wards/treatment areas in most hospitals. However, none of the 13 hospitals conducted a systematic audit of the presence of posters or provided other forms of hand hygiene promotion materials, including hand hygiene stickers or badges. Only three hospitals had other forms of hand hygiene posters. The performance of five secondary hospitals was rated good in this domain whereas four of the tertiary hospitals were rated very poor.

Institutional Safety Climate: The institutional safety climate for hand hygiene assesses the institutional culture of teamwork, leadership support, patient engagement, and designation of hand hygiene role models or champions.

Twelve of the hospitals had dedicated hand hygiene teams. Of these, 10 had regular meetings and seven dedicated time to promote hand hygiene in their facilities. Systems for the designation of hand hygiene champions, and recognition and utilization of hand hygiene role models were available in seven and 10 hospitals, respectively. However, a formalized patient engagement program was available in only three hospitals. Five tertiary hospitals and two secondary hospitals were rated poor in this domain, and only one hospital (tertiary hospital) was rated excellent in this domain.

## 4. Discussion

To our knowledge, this study provides the first publication on hand hygiene practices and promotion activities using the WHO HHSAF tool in a number of public hospitals in West Africa. This evidence has important policy implications for hand hygiene practice as it provides information on its implementation in public hospitals in the most populous region of Sierra Leone.

The mean hand hygiene level reported for the public hospitals in Sierra Leone (HHSAF score = 273) was lower than the global mean hand hygiene levels reported by WHO in 2011 (HHSAF score = 335) and 2015 (HHSAF score = 374) and that reported in the United States of America in 2011 (HHSAF score = 373) [19,20]. We were not able to find a comparable score in the global hand hygiene survey across LMICs where there are many barriers to successful hand hygiene practice, including limited knowledge and awareness of hand hygiene among healthcare workers and a lack of resources to practice hand hygiene [21,22,23,24]. These barriers may have resulted in lower mean scores for hand hygiene being reported from Sierra Leone and other LMICs.

The mean level of hand hygiene in our study was higher than that reported in Cambodia (HHSAF score = 178), India (HHSAF score = 225), and Tanzania (HHSAF score = 187) [25,26,27]. Unlike the health facilities in Cambodia, in which no hospital achieved an intermediate hand hygiene level [25], nine of the 13 hospitals in our study achieved an intermediate hand hygiene level. Sierra Leone is one of the countries in West Africa that has had high-risk infectious disease outbreaks, including the 2014/2016 Ebola virus disease epidemic [28]. This resulted in the establishment of the National IPC program in 2015 and a series of interventions on IPC [13]. Since the first confirmed case of COVID-19 was reported on 31 March 2020, the Sierra Leone government has again stepped up many hand hygiene measures, including training and provision of resources [29,30]. Despite these achievements, more effort is needed to improve and sustain the practice of hand hygiene as none of the hospitals in the Western Area of Sierra Leone was able to reach the advanced hand hygiene level and thus qualify for the leadership assessment, in contrast to 45% of hospitals in the United States of America (USA) [20] and 60% of hospitals in South Korea [31].

We have identified gaps that should be addressed to ensure further improvement and progress on hand hygiene promotion activities in Sierra Leone. First, there are gaps in the provision of continuous supplies of soap and running water, as well as the availability of single-use towels for hand drying, similar to Cambodia [25]. This is due to a lack of practical plans and dedicated budgets to improve hand hygiene infrastructure and maintain these supplies. Therefore, support from the government and development partners is needed to improve the hand hygiene infrastructure in the country’s public hospitals.

A major gap in hand hygiene education and training is the lack of dedicated budgets to support training services in public hospitals. As many public hospitals are severely underfunded and unable to carry out many normal facility operations, the hospital administrators may need to explore other sources of funding for sustaining local initiatives on hand hygiene practices and promotion activities.

There were three tertiary hospitals and two secondary hospitals that were rated poor on evaluating their hand hygiene practices and promotion. The main gaps in the evaluation of hand hygiene promotion activities were lack of an audit system of hygiene resources and poor compliance with hand hygiene practices. This is not unique to this study, as a recent study reported a low compliance rate of healthcare workers on hand hygiene in Sierra Leone [21].

In the workplace reminders domain, a major gap is the lack of a system to review the positioning of posters or other resources needed for hand hygiene advocacy. Four tertiary hospitals and three secondary hospitals reported low to very low scores for hand hygiene reminders in the workplace. Posters on hand hygiene at various places in hospitals, especially in hand hygiene stations, intensive care units or operating theatres can be very important reminders as reported in Nigeria and Rwanda [32,33].

Of all the domains of the HHSAF tool, the institutional safety climate for hand hygiene scored lowest, especially in secondary hospitals, similar to a report from Cambodia [23]. The key challenge related to the institutional safety climate is the lack of a formalized patient engagement program and the lack of a system for the designation of hand hygiene champions or role models in many of the hospitals. Therefore, hand hygiene promotion activities should be integrated into the culture of hospitals and communities.

The mean HHSAF score for tertiary hospitals was higher than for secondary hospitals, but this was not statistically significant. This may be due to the lack of dedicated budgets and practical plans to improve hand hygiene infrastructure in secondary hospitals. In general, however, there was a strong similarity in the distribution of scores across domains of HHSAF between secondary and tertiary hospitals. This observation may be due to the fact that the hospitals have similar sources of funding (mostly from the government of Sierra Leone). Overall, though the hospitals in this study achieved an intermediate level of hand hygiene practices and promotion, there were variations in the scores across domains, with several hospitals being rated poor in certain domains. This calls for the attention of the hospital administrators and policy makers in the country’s health ministry.

The strengths of our study were that we assessed hand hygiene policy using a well-validated WHO tool in a typical African setting and obtained data from all the hospitals we approached. The health ministry has been informed about the study, and other key stakeholders such as WHO are engaged in the study.

Our study also has some limitations. First, the study was conducted only in public hospitals in the Western Area of Sierra Leone and therefore did not provide information on the level of hand hygiene in private and provincial hospitals. As the HHSAF is a self-assessment tool, some of the participating facilities may not have given the correct picture of hand hygiene implementation in their facilities. This is because some people may want to project a clean picture of their facility, or rather, some people may want to create a picture to attract more hand hygiene resources to their facility. We were only able to access the data for one year, so we do not know if the scores have improved over time or not.

## 5. Conclusions

The implementation of hand hygiene practices and promotion in 13 public hospitals is mostly at an intermediate level, with variation in performance across the domains. Many hospitals were rated poor in institutional safety climate and reminders in the work pace domains, but most excelled in training and education. This inadequate practice and promotion of hand hygiene in the country is likely to increase the spread of HAIs. Long-term sustainable planning is needed to improve hand hygiene practice and promotion by enhancing training, facilitating the distribution of hand hygiene resources within hospitals, and encouraging an embedded culture of hand hygiene in hospitals and communities. This study did not examine factors influencing the implementation of hand hygiene services, so we recommend further research to understand barriers and guide hospital administrators and policymakers.

## Figures and Tables

**Figure 1 ijerph-19-03787-f001:**
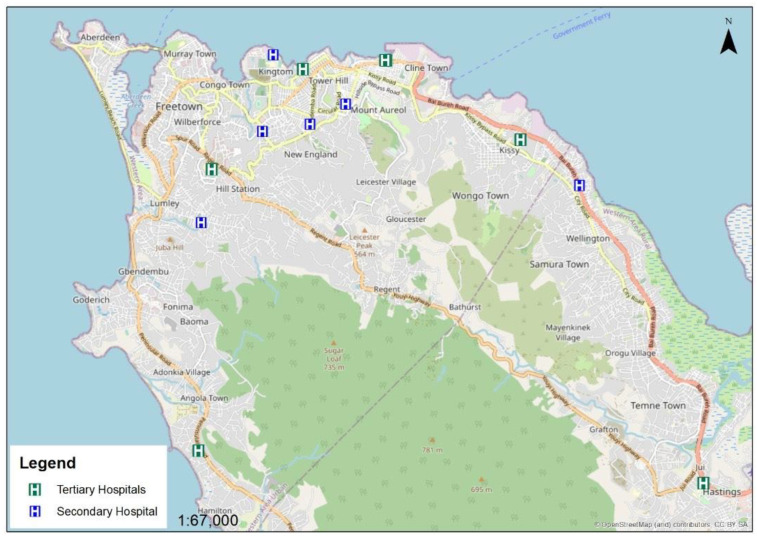
Map of Freetown showing the 13 public hospitals in the Western Area of Sierra Leone.

**Table 1 ijerph-19-03787-t001:** Hand hygiene levels as assessed by scores from the questions in the WHO hand hygiene self-assessment framework.

Total Score	Hand Hygiene Level	Definition
0–125	Inadequate	indicates insufficient hand hygiene practices and promotion, and requires significant improvement
126–250	Basic	indicates that some measures are in place but not satisfactory and therefore requires further improvement
251–375	Intermediate or Consolidation	indicates appropriate hand hygiene promotion strategies and improvements in hand hygiene practices, but requires long-term planning to ensure continual improvement and progress
376–500	Advanced or Embeddinge	iindicates sustained hand hygiene promotion and practice as well as a quality and safety culture surrounding hand hygiene promotion within the organization

**Table 2 ijerph-19-03787-t002:** Characteristics of the 13 hospitals in the Western Area of Sierra Leone assessed on hand hygiene policy using the HHSAF in 2021.

Hospital Characteristics	Secondary Hospitals (N = 6)	Tertiary Hospitals (N = 7)
	**N (%)**	**N (%)**
**Type of hospital**	6 (100)	7 (100)
**Location of hospital**		
Urban	6 (100)	5 (71)
Rural	0 (0)	2 (29)
**Bed capacity**		
<50	3 (50)	0(0)
51–100	3 (50)	1 (14)
101–150	0 (0)	1 (14)
151–200	0 (0)	2 (29)
>200	0 (0)	3 (43)
**Staff capacity**		
≤200	2 (33)	3 (43)
201–400	4 (67)	0 (0)
400–600	0 (0)	3 (43)
≥601	0 (0)	1 (14)
**Units/wards**		
<10	1 (17)	0 (0)
10–20	4 (66)	4 (57)
>20	1 (17)	3 (43)

**Table 3 ijerph-19-03787-t003:** Categories of healthcare workers in 13 public hospitals in the Western Area of Sierra Leone assessed on hand hygiene policy using the HHSAF in 2021.

Hospital Type	Hospital	Nurses	Doctors & CHOs ^†^	Pharmacy & Laboratory Personnel	Others *
**Secondary**	S1	149	10	11	33
	S2	124	12	3	6
	S3	252	15	29	16
	S4	173	20	30	51
	S5	184	6	19	14
	S6	35	1	5	2
**Tertiary**	T1	420	36	24	87
	T2	295	31	23	34
	T3	51	5	14	35
	T4	62	9	8	13
	T5	87	9	4	3
	T6	294	49	25	79
	T7	442	42	25	137

* Others include hospital administrators, radiographers, hygienists, and porters; † CHOs: Community Health Officers.

**Table 4 ijerph-19-03787-t004:** The 2021 HHSAF assessment scores of 13 public hospitals in the Western Area of Sierra Leone.

Hospital Type	Hospital/Mean ± SD	SC	TE	EF	RW	ISC	Total Score	Hand Hygiene Level
**Secondary**	S1	35	80	48	55	50	268	Intermediate
	S2	40	35	58	50	30	248	Basic
	S3	55	80	65	50	65	315	Intermediate
	S4	80	50	45	70	30	275	Intermediate
	S5	50	70	60	45	65	290	Intermediate
	S6	75	50	68	63	50	305	Intermediate
	**Mean ± SD**	**48 ± 11**	**72 ± 21**	**55 ± 14**	**36 ± 15**	**43 ± 13**	**260 ± 30**	**Intermediate**
**Tertiary**	T1	40	100	75	33	35	283	Intermediate
	T2	65	75	65	25	65	295	Intermediate
	T3	55	65	35	15	40	210	Basic
	T4	50	75	53	38	40	255	Intermediate
	T5	30	55	35	68	25	213	Basic
	T6	30	55	35	68	25	213	Basic
	T7	75	80	60	70	90	375	Intermediate
	**Mean ± SD**	**56 ± 21**	**63 ± 13**	**53 ± 14**	**62 ± 10**	**50 ± 25**	**284 ± 58**	**Intermediate**

SC = system change; TE = training and education; EF = evaluation and feedback; RW = reminders in the workplace; ISC = institutional safety climate; S1 to S6: secondary hospitals 1 to 6 and T1 to T7: Tertiary hospitals 1 to 7; Green = excellent performance (>70) Yellow = good performance (50–70); Orange = poor performance (35–50) Red = very poor performance (<35).

## Data Availability

Data is available at the University of Sierra Leone repository and will be available on request.

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
