# Peer review of "How Well Are Hand Hygiene Practices and Promotion Implemented in Sierra Leone? A Cross-Sectional Study in 13 Public Hospitals"

_ijerph, 2022, doi:10.3390/ijerph19073787_

Round 1
Reviewer 1 Report
General comment
The present manuscript was focused on evaluation of hand hygiene practices and promotion across 13 public 31 health hospitals in the Western Area of Sierra Leone in 32 a cross-sectional study using the WHO Hand Hygiene Self-Assessment Framework. This work is significant due to the importance of hand hygiene, especially in countries where hygiene measures can be underdeveloped. Results are useful to fill the gap concerning this aspect in Sierra Leone as well as in other countries with the low number of hygiene implementation programs. This paper can be accepted, but it presents some drawbacks as listed in the following section.
Specific comments
Line 52: each abbreviation must be given in full for the first time (LMICs)
Line 78-84: to my opinion, these lines should be moved to discussion section.
Line 115: what type of interview was carried out? Authors need to be more accurate in this description because it represents the focus of this methodology.
Line 129: data collection is not clear. Please, clarify the collection based on paper-approach. Furthermore, in box 1 the total score for each aspect has a range and a better explanation of a good explanation of how the values ​​in this range were decided would be appropriate.
Box 1 must be separated by the text of manuscript (line 128)
Line 153 and 168: remove dots between 3.2.1 and the total facility…
Line 164-167: footnotes must be reported with the same font
Author Response
|
Reviewer 1 General comment The present manuscript was focused on evaluation of hand hygiene practices and promotion across 13 public 31 health hospitals in the Western Area of Sierra Leone in 32 a cross-sectional study using the WHO Hand Hygiene Self-Assessment Framework. This work is significant due to the importance of hand hygiene, especially in countries where hygiene measures can be underdeveloped. Results are useful to fill the gap concerning this aspect in Sierra Leone as well as in other countries with the low number of hygiene implementation programs. This paper can be accepted, but it presents some drawbacks as listed in the following section. Response: Thank you |
|
Specific comments
Response: We have rectified this.
Response: Thank you for this observation. However, we felt the details in lines 78-84 which state that “In 2014, Sierra Leone faced an Ebola outbreak that infected over 14,000 people, killed 3,589 people, and adversely affected the operations of the health system [13]. As a result of this, the Ministry of Health established the National Infection Prevention and Control (IPC) program in 2015 for the first time and developed an IPC policy to guide the implementation of hand hygiene [14]. Since then, the National IPC Program has used the WHO HHSAF tool annually to collect data on the status of hand hygiene promotion activities within healthcare facilities. However, this data has not been routinely analyzed” provide the audience information on the progress in the implementation of Infection Prevention and Control activities in Sierra Leone. Furthermore, these details highlight the challenges in the assessment of hand hygiene practices and promotion in Sierra Leone using the HHSAF tool. We believe that providing this information in context will provide a clearer picture of hand hygiene and IPC practices in Sierra Leone. Line 115: what type of interview was carried out? Authors need to be more accurate in this description because it represents the focus of this methodology. Response: We used a structured interview approach to obtain information from the IPC focal person using the paper format of the HHSAF tool. We have corrected this in the paper (lines 48 to 50).
Response: As stated above, we used the paper format of the HHSAF tool to collect the data from the IPC focal persons. We have rectified this in the paper (lines 48-50). Furthermore, in box 1 the total score for each aspect has a range and a better explanation of a good explanation of how the values ​​in this range were decided would be appropriate. Response: The range of scores highlighted in Box 1 is derived from the hand self-assessment framework tool designed by the World Health Organization. These are not arbitrary values, but values obtained from the Hand Hygiene Self-Assessment Framework tool. Twenty-seven indicators were developed as binary or multiple-choice questions designed to facilitate the assessment of hand hygiene promotion. We have amended the title of box 1 to make this clearer.
Response: Thank you. We rectified this
Response: We have corrected this (lines 205, 234, and 242)
Response: We have corrected this (lines 205 to 209)
|
Reviewer 2 Report
The study assessed how well hand hygiene practices and promotion are implemented in 13 public hospitals in Sierra Leone using the WHO Hand Hygiene Self-Assessment Framework. The study is well-conceived and presented. The discussions are well supported by the results.
However, minor edit for readership clarity is appended in the attached.
Besides, possible questions derivable from the research may include; What is the state or level of hand hygiene practices and promotion activities in 13 public health hospitals in the Western Area of Sierra Leone? What are the challenges to implementing hand hygiene practices and promotion activities?
The has widely been researched in the West African region where Sierra Leone is situated. Nevertheless, the research is relevant in the field given the Sierra Leone context
The research only supports the known facts given previous knowledge from existing publications within the region
A statistical correlation between hand hygiene practices and length of stay in hospital could be a useful addition to the research
The conclusions are moderate. They could be improved will be validated statistical analysis such as suggested above The references are appropriate in my cases of citation
Some additional comments on the tables and figures: A study area figure could be needed, showing the distribution/location of the 13 hospitals

Author Response
Reviewer 2
Comments and Suggestions for Authors
The study assessed how well hand hygiene practices and promotion are implemented in 13 public hospitals in Sierra Leone using the WHO Hand Hygiene Self-Assessment Framework. The study is well-conceived and presented. The discussions are well supported by the results.
Response: Thank you
However, minor edit for readership clarity is appended in the attached.
- Besides, possible questions derivable from the research may include; What is the state or level of hand hygiene practices and promotion activities in 13 public health hospitals in the Western Area of Sierra Leone? What are the challenges to implementing hand hygiene practices and promotion activities?
Response: We agree with your observation. We realize this is a gap that has not been addressed by the evidence provided to readers in this paper. As such, we stated in the conclusion of the manuscript that ‘This study did not examine factors influencing the implementation of hand hygiene services, so we recommend further research to understand barriers and guide hospital administrators and policymakers.
- The has widely been researched in the West African region where Sierra Leone is situated. Nevertheless, the research is relevant in the field given the Sierra Leone context
Response: Thank you
- The research only supports the known facts given previous knowledge from existing publications within the region
Response: We agree with this
- A statistical correlation between hand hygiene practices and length of stay in hospital could be a useful addition to the research
Response: We agree with this but assessing the length of hospital stay is beyond the scope of this work.
- The conclusions are moderate. They could be improved will be validated statistical analysis such as suggested.
Response: Thank you
- above The references are appropriate in my cases of citation
Response: Thank you
- Some additional comments on the tables and figures: A study area figure could be needed, showing the distribution/location of the 13 hospitals
Response: We have added this in the paper (Figure 1, lines 130 to 133)
- It could be interesting to know the number of ward or unit in the 13 hospitals facilities to help understand the total sample size or participants
Response: We have added details on the number of units or wards of the 13 hospitals in table 1. However, the study did not assess hand hygiene promotion activities in specific wards or units, as the hand hygiene self-assessment framework was designed for the assessment of hand hygiene promotion activities at the facility level.
Reviewer 3 Report
Authors used a descriptive study to identify the situation of hand hygiene at public hospitals in Sierra Leone. Because hand hygiene is one of the most important practices for preventing healthcare-associated infections, it may be meaningful to show the situation of hand hygiene in Sierra Leone. However, this article has not been fully answered some of questions due to the insufficient description.
First, authors suggested ““the study” interviewed the ICP focal persons…”, but they did not show who interviewed the IPC focal point. Interviewers may sometimes affect interviewees, which may change the answer. Authors should disclose who interviewed the ICP persons in the method section.
Second, authors compare the differences between secondary hospitals and tertiary hospitals in table 1, table 2 and table 3, but they did not examine statistical differences (i.e. p-values). The sample size is small, but authors should show the p-values between secondary hospitals and tertiary hospitals in table 1, table 2 and table 3 as well as the results section.
Finally, authors compared the differences between secondary hospitals and tertiary hospitals in the method section, but there is no description of discussion for the differences in the discussion section. Authors should add the discussion about the differences in the discussion section.
Minor comments
- Authors use comma ”,” for decimal point, but they should use period “.”. (e.g., L35, L36, L158, L160).
Author Response
Reviewer 3
Comments and Suggestions for Authors
Authors used a descriptive study to identify the situation of hand hygiene at public hospitals in Sierra Leone. Because hand hygiene is one of the most important practices for preventing healthcare-associated infections, it may be meaningful to show the situation of hand hygiene in Sierra Leone. However, this article has not been fully answered some of questions due to the insufficient description.
- First, authors suggested ““the study” interviewed the ICP focal persons…”, but they did not show who interviewed the IPC focal point. Interviewers may sometimes affect interviewees, which may change the answer. Authors should disclose who interviewed the ICP persons in the method section.
Response: The Principal Investigator with support from three research assistants interviewed the IPC focal persons. We have rectified this in the manuscript (Lines 48-50).
- Second, authors compare the differences between secondary hospitals and tertiary hospitals in table 1, table 2 and table 3, but they did not examine statistical differences (i.e. p-values). The sample size is small, but authors should show the p-values between secondary hospitals and tertiary hospitals in table 1, table 2 and table 3 as well as the results section.
Response: We agree that statistical differences between secondary and tertiary hospitals need to be examined, but due to the small sample size, it has no statistical significance.
Results:
- Finally, authors compared the differences between secondary hospitals and tertiary hospitals in the method section, but there is no description of discussion for the differences in the discussion section. Authors should add the discussion about the differences in the discussion section.
Response: We have added increased details on this in the paper. (lines 354 to 359)
Minor comments
- Authors use comma”,” for decimal point, but they should use period “.”. (e.g., L35, L36, L158, L160).
Response: Thank you. We have corrected this in lines 41 and 42 and lines 206 to 212
Round 2
Reviewer 3 Report
I have no further comment to this manuscript.